# Dose-dependent effects of rumen-protected choline on hepatic metabolism during induction of fatty liver in dry pregnant dairy cows

**Usman Arshad**[1], **Marcos G. Zenobi**[1], **Paula Tribulo**[1], **Charles R. Staples**[1†], **José E. P. Santos**[1,2]*

**1** Department of Animal Sciences, University of Florida, Gainesville, FL, United States of America, **2** DH Barron Reproductive and Perinatal Biology Research Program, University of Florida, Gainesville, FL, United States of America

† Deceased.
* jepsantos@ufl.edu

**Data Availability Statement:** We created repository files in which the data base part of this experiment have been uploaded and is in the public domain in FigShare. We also created a repository

## Abstract

Objectives were to determine the effects of supplementing increasing amounts of choline ion on hepatic composition and mRNA abundance in pregnant dry cows subjected to a fatty liver induction protocol. Holstein cows (35 primiparous and 41 multiparous) at mean (± standard deviation) of 211 ± 9.9 days of gestation were blocked by body condition (3.59 ± 0.33) and assigned to receive 0, 6.45, 12.90, 19.35, and 25.80 g/day of choline ion as rumen-protected choline (RPC) as a top-dress for 14 days. Cows were fed for ad libitum intake on days 1 to 5 and restricted to 30% of the required net energy for lactation from days 6 to 14 of the experiment. Hepatic tissue was sampled on days 5 and 14 and analyzed for concentrations of triacylglycerol and glycogen, and mRNA abundance was investigated. Orthogonal contrasts evaluated the effects of supplementing RPC (0 g/day vs. rest), and the linear, quadratic, and cubic effects of increasing intake of choline ion from 6.45 to 25.80 g/day. Results are depicted in sequence of treatments from 0 to 25.8. During feed restriction, RPC reduced the concentration of hepatic triacylglycerol by 28.5% and increased that of glycogen by 26.1%, and the effect of increasing RPC intake on triacylglycerol was linear (6.67 vs. 5.45 vs. 4.68 vs. 5.13 vs. 3.81 ± 0.92% wet-basis). Feeding RPC during feed restriction increased abundance of transcripts involved in choline metabolism (*CHKA*, *PLD1*), synthesis of apolipoprotein-B100 (*APOB100*), and antioxidant activity (*GPX3*), and decreased the abundance of transcripts involved in hepatic lipogenesis (*DGAT2*, *SREBF1*) and acute phase response (*SAA3*). Most effects were linear with amount of choline fed. Changes in hepatic mRNA abundance followed a pattern of reduced lipogenesis and enhanced lipids export, which help explain the reduced hepatic triacylglycerol content in cows fed RPC. Choline exerts lipotropic effects in dairy cows by altering transcript pathways linked to hepatic lipids metabolism.

file for the supplemental Tables. Data Link: https://figshare.com/s/c4a24d1373a8f9962191
Supplemental Tables: https://figshare.com/s/ed3415a393c2b4dbf3a8.

**Funding:** The Higher Education Commission of Pakistan partially funded the Ph.D. program of the first author, Usman Arshad. The funder had no role in study design, data collection and analysis, decision to publish, or preparation of the manuscript.

**Competing interests:** The authors have declared that no competing interests exist.

## Introduction

Dairy cows undergo periods of extensive negative nutrient balance in the last week of gestation and first 4 to 6 weeks of lactation that increases lipomobilization, thus resulting in an increased concentrations of free fatty acids in blood [1]. Long-chain fatty acids in blood are taken up by the hepatic tissue and undergo multiple fates such as complete synthesis of other lipid products, oxidation, ketogenesis, or re-esterification to triacylglycerol, which eventually are exported as lipoproteins; however, extensive re-esterification of fatty acids in hepatocytes results in hepatic lipidosis [1,2]. Theinert et al. [3] collected hepatic tissue from Holstein cows 14 days before and 7 days after calving and showed that concentrations of triacylglycerol increased from 2.5 to 15.0% on a tissue wet-basis. Accumulation of excessive concentrations of triacylglycerol in hepatic tissue is associated with impaired production, health, and survival in dairy cows [4].

Choline needs to be fed in rumen-protected forms known as rumen-protected choline (**RPC**) to ruminants because microbial activity in the forestomachs results in almost complete degradation of dietary choline [5]. Supplementation of RPC has been shown to reduce triacylglycerol deposition into the hepatic tissue [6], increase choline-derived phospholipids in blood [7], and improve productive performance in dairy cows [8]. Presumably, the benefits observed when choline is fed to dairy cows involve lipotropic actions in the liver. Goselink et al. [9] supplemented 12.9 g/day of choline ion as RPC to transition cows and investigated the effects on transcripts involved in hepatic lipid metabolism. The authors observed that mRNA abundance of microsomal triglyceride transport protein (**MTTP**) and apolipoprotein-B100 (**APOB100**), both involved in the assembly of very-low density lipoprotein (**VLDL**) particles, increased in hepatic tissue collected on weeks 1, 3, and 6 postpartum in cows fed RPC. It is well known that VLDL particles are important carriers of triacylglycerol in blood and critical for hepatic export of lipids [10]; nevertheless, their assembly and secretion require phosphatidylcholine [11]. Artegoitia et al. [12] showed that concentrations of choline metabolites in early lactation are reduced, and transition dairy cows affected by fatty liver display smaller concentrations of phospholipids in plasma than cows without fatty liver [13]. Thus, supplementing choline ion as RPC likely increases substrate availability for synthesis of phosphatidylcholine; nevertheless, experiments manipulating the amounts of choline ion to elucidate their effects on hepatic metabolism when cows are experimentally induced to develop fatty liver are scarce.

When cows undergo extensive lipomobilization, the lipids deposited into the hepatic tissue can undergo peroxidation [14]. Such changes can compromise the ability of hepatocytes to dispose of fatty acids, thus further increasing the severity of hepatic lipidosis [15]. Bradford et al. [16] pair-fed control cows with those receiving recombinant bovine tumor-necrosis factor-alpha (**TNFA**) such that they would have the same dry matter intake (**DMI**). They showed that TNFA increased hepatic accumulation of triacylglycerol independent of DMI; therefore, nutritional strategies that reduce inflammation might alleviate the risk of fatty liver in dairy cows. Phosphatidylcholine has anti-inflammatory properties [17], and it is plausible that if feeding RPC increases synthesis of phospholipids, it might affect molecular pathways involved in inflammation or oxidative damage that might affect the severity of fatty liver in dairy cows.

We hypothesized that supplementing choline ion as RPC alters transcript pathways in the hepatic tissue involved in export of triacylglycerol and that have anti-inflammatory effects. Such changes in hepatic tissue transcript abundance underlies the potential reduction in hepatic triacylglycerol content when cows are subjected to develop fatty liver. Therefore, the objectives of this experiment were to determine the effect of increasing amounts of choline ion fed as RPC on hepatic tissue abundance of transcripts linked to numerous pathways that favor export of triacylglycerol and minimize inflammation and cellular oxidative damage in dairy cows.

## Materials and methods

The experiment was conducted at the University of Florida Dairy Unit (Hague, FL) from December 2015 to June 2016 and all procedures with cows were approved by the Institutional Animal Care and Use committee of the University of Florida protocol number 201509174. This manuscript is a companion paper from an experiment [7]. Details of cows, housing and feeding management, dietary ingredient sampling and chemical analysis, body condition score (**BCS**) and body weight (**BW**) are presented in Zenobi et al. [7]. The diets offered during the ad libitum and feed restriction periods and the estimated supply of nutrients are provided in **S1 Table.**

### Experimental design, dietary treatments, cows, and housing

Seventy-six pregnant, non-lactating Holstein cows, 35 that just completed their first lactation and 41 that completed 2 or more lactations, with a mean (± standard deviation) of 211 ± 9.9 days of gestation and a BCS of 3.59 ± 0.33 were enrolled in the experiment. The cows were housed in a tie-stall facility and fed ad libitum during the first 5 days, whereas feed restriction was imposed from days 6 to 14 of the experiment. The experiment followed a randomized complete block design. Every 2 weeks, a cohort of 7 to 10 cows that had recently been dried off were enrolled in the experiment. Within a cohort, cows were ranked by BCS, from the smallest to the largest value, and then each 5 cows were assigned a block. Within each block, cows were assigned randomly to receive increasing amounts of choline ion from a RPC source containing 28.8% choline chloride (ReaShure, Balchem Corp., New Hampton, NY). The five treatments were **0** (0 g/day of choline ion as RPC; n = 15 cows); **6.5** (6.45 g/day of choline ion from 30 g/day ReaShure; n = 15); **12.9** (12.90 g/day of choline ion from 60 g/day ReaShure; n = 14); **19.4** (19.35 g/day of choline ion from 90 g/day ReaShure; n = 15); and **25.8** (25.80 g/day of choline ion from 120 g/day ReaShure; n = 17). The specific amounts of RPC for each treatment were mixed with 75 g of ground corn, 25 g of dried molasses, and 14 g of sodium chloride and top-dressed daily to each cow during the 14-day experiment. Thus, the amount of top-dress fed daily to cows in each treatment differed, increasing from 114 g/day for those receiving the treatment with 0 g/day of choline ion up to 234 g/day for cows treated with 25.8 g/day of choline ion.

### Ad libitum feeding period

During the ad libitum period, dry Holstein cows were fed a diet designed according to NRC [18] from days 1 to 5 of the experiment. It was anticipated that they would consume 12 kg dry matter per day to achieve a positive net energy for lactation (**$NE_L$**) balance. The amounts of $NE_L$ and amino acids supplied by the diets were calculated using the observed DMI and the nutrient composition of the diets based on the chemical analyses of the ingredients in NASEM [19]. The estimated values resulted in a diet with $NE_L$ content of 1.61 Mcal/kg that resulted in slight negative energy balance because DMI was less than initially anticipated according to NASEM [19]. The supply of metabolizable protein was 925 g/day with 20 g/day of metabolizable methionine estimated using NASEM [19]. The NASEM publication [19] uses an estimated efficiency of use of metabolizable protein of 0.69, except for incorporation into the pregnant uterus (efficiency of 0.33), or excretion of urinary nitrogen (efficiency of 1). Cows were fed once daily at 0900 hours, and the amounts of feed offered to individual cows were adjusted once daily to ensure at least 5% refusals, which were weighed before the morning feeding.

### Feed restriction period

During the feed restriction period, from experimental days 6 to 14, the diet was formulated and offered in amounts to supply approximately 31% of the $NE_L$ required for maintenance

and pregnancy according to NRC [18]. The mean BW and gestation length of each block of 5 cows was used to calculate the metabolic BW with the associated maintenance requirement and the gestation length which were used to generate the pregnancy needs using an expected 42 kg of BW for the calf at birth. Based on the NASEM [19], the imposed feed restriction resulted in a supply of $NE_L$ representing 30% of the daily required energy. During the feed restriction period, cows consumed mean (± standard deviation) 3.31 ± 0.26 kg/d of DMI or 0.48 ± 0.03% of BW. The calculated values resulted in a diet with $NE_L$ content of 1.48 Mcal/kg and a supply of metabolizable protein of 235 g/day with 20 g/day of metabolizable methionine with the efficiency of metabolizable protein use as described in the ad libitum feeding period according to NASEM [19].

Feed restriction was used to induce lipomobilization and increase accumulation of triacylglycerol into hepatic tissue as previously reported [6]. The diet was formulated including rumen-protected methionine (Smartamine M, Adisseo USA, Alpharetta, GA) to supply a similar amount of metabolizable methionine as in the ad libitum feeding period [7]. The mineral vitamin premix comprised a greater proportion of the diet during the feed restriction period compared with the ad libitum period to supply similar amounts of micronutrients and ionophore [7].

## Hepatic tissue collection and processing

Hepatic tissue was collected on days 5 and 14 of the experiment and details of the procedures for quantification of triacylglycerol, glycogen, and DNA contents are described elsewhere [4,7].

Total RNA was extracted from hepatic tissue sampled on days 5 and 14 using Trizol (TRIzol LS Reagent, Invitrogen, Waltham, MA). In summary, approximately 25 mg of hepatic tissue were added to a microtube containing zirconium oxide beads (CKMix, Bertin Corp., Thermo Fisher Scientific, Waltham, MA) containing 800 μL of Trizol. Chloroform was added to bring up to a 20% chloroform solution. The tubes were homogenized vigorously by hand for 15 s and incubated at room temperature for 3 min. Samples were homogenized 3 times for 20 s at 6,200 rpm using a homogenizer (Precellys 24, Bertin Corp. Thermo Fisher Scientific). Samples were then centrifuged at $12,000 \times g$ for 15 min at 4°C, and the colorless supernatant containing the RNA was transferred to a new microtube. Purification of RNA was performed using the Quick-RNA 96 kit (Zymo Research, Irvine, CA) according to the manufacturer's instruction. Purity and concentration were evaluated using a NanoDrop 2000 spectrophotometer (Thermo Fisher Scientific). Samples on days 5 and 14 had a mean (± standard deviation) 260:280 nm ratio of 2.25 ± 0.22 (range 1.98 to 2.85), and 2.16 ± 0.11 (range 1.99 to 2.42), respectively.

The mRNA for a selected set of transcripts was quantified by the Fluidigm Biomark HD quantitative PCR microfluidic system (Fluidigm Co., San Francisco, CA). The PCR primers were designed by Fluidigm Delta Gene assays and synthesized by Fluidigm (Fluidigm Co.). Details of transcripts and primers are presented in **S2 Table.** A pooled sample containing mRNA from bovine hepatic tissue from 10 different samples was used for primer validation. Primers were validated using Fluidigm primer quality-control criteria ($R^2 \geq 0.97$; efficiency of 97.6 to 130.3%; slope = −3.92 to −2.76) were applied to cDNA serially diluted by 12× and evaluated in 8 replicates. Reference genes used were *ACTB*, *RPL12*, and *RPS9*.

Fluidigm panel was constructed to investigate transcripts of interest which are involved in metabolism of choline (*BHMT*, *CBS*, *CEPT1*, *CHDH*, *CHKA*, *CHPT1*, *GAMT*, *GNMT*, *MAT1A*, *MTHFR*, *MTR*, *PCYT1A*, *PCYT1B*, *PEMT*, *PLD1*), insulin biosynthesis (*PCSK1*), fibroblast growth factor (*FGF21*), glucose metabolism (*G6PC*, *PCK1*, *PCK2*, *SLC2A1*, *SLC2A2*), fatty acids trafficking (*CD36*, *FABP1*, *SLC27A1*, *SLC27A2*, *SLC27A5*), carnitine metabolism

(*BBOX1*, *CRAT*, *CROT*, *SLC22A5*, *TMLHE*), fatty acids activation and oxidation (*ACADM*, *ACSL1*, *CPT1A*, *PPARA*, *PPARG*, *SLC25A20*), synthesis or re-esterification of fatty acids (*ACACA*, *ACLY*, *FASN*, *DGAT2*), phospholipids metabolism (*AGPAT2*, *AGPAT3*, *PLA2G1B*, *PTTDS1*, *SGMS1*, *SMPD1*), synthesis and assembly of VLDL particles (*APOB100*, *APOE*, *HDLBP*, *MTTP*, *PCTP*, *VLDLR*), cholesterol metabolism (*ABCA1*, *ABCG1*, *CYP7A1*, *FDPS*, *MVK*), cell signaling (*NR1H3)*, de novo hepatic lipogenesis (*SREBF1*, *SREBF2*), ketogenesis (*ACAT1*, *HMGCL3*, *HMGCR*, *HMGCS1*), cytokines (*IL1B*, *IL10*, *MGST3*, *TNFA*), acute phase response (*CRP*, *HP*, *SAA3*), oxidative damage (*HMOX2*), synthesis of antioxidants (*CAT*, *GPX3*, *MT1A*, *MT1E*, *MT2A*, *NQO1*, *SOD1*) pathogen recognition receptor (*TLR4*), lipid per-oxidation (*ALDH7A1*, *ALDH9A1*), glycerol phosphate pathway (*GDE1*, *GDPD1*, *GDPD2*, *GPAM*), urea cycle (*ASS1*, *OTC*), glucuronidation reaction (*UGT1A1*), and endoplasmic retic-ulum stress response (*XBP1*).

Primers targeting 18 transcripts (*ACLY*, *CHDH*, *FASN*, *GDPD1*, *GDPD2*, *PCSK1*, *PCYT1B*, *PLA2G1B*, *PPARG*, *SGMS1*, *SLC2A1*, *SLC22A5*, *SLC27A1*, *SLC27A5*, *SREBF2*, *TLR4*, *TNFA*, and *VLDLR*) failed to pass the quality control in the qualification run and were excluded from the analyses. Primer efficiency in the transcripts passing quality control and used in this experi-ment ranged between 97.6 and 130.3% (**S2 Table**). The geometric mean of the cycle threshold (**Ct**) of all three reference genes was calculated for each sample. None of the reference transcript was affected by treatments in hepatic tissues collected on day 5; however, *RPS9*, a reference gene was affected by treatments in hepatic tissues collected on day 14 and not included to calculate the geometric mean of Ct references genes. Statistical analyses were performed on the delta Ct (**dCt**) values as described by Steibel et al. [20]. Fold changes relative to CON were calculated using the method described by Yuan et al. [21], whereby fold changes were calculated from least squares means (**LSM**) difference or delta-delta Ct (**ddCt**) according to the formula $2^{-ddCt}$, where dCt = CtTarget gene–geometric mean of CtReference genes, and ddCt = dCtTreatment A–dCtTreatment B. Heatmaps were generated using Heatmapper online tool [22].

## Statistical analyses

Normality of residuals and homogeneity of variance were examined for each continuous dependent variable analyzed after fitting the statistical model. Responses that violated the assumptions of normality were subjected to power transformation according to the BoxCox procedure [23] using a macro [24] for mixed models in SAS (SAS/STAT, SAS Institute Inc., Cary, NC). The LSM and respective standard errors of the means (**SEM**) were back trans-formed for presentation of results according to Jørgensen and Pedersen [25].

Data were analyzed by ANOVA with linear mixed models using the MIXED procedure of SAS, and analyses were performed separately for the ad libitum and feed restriction periods. The statistical models included the fixed effects of treatment (0 vs. 6.5 vs. 12.9 vs. 19.4 vs. 25.8), the BCS and the BW at the time of enrollment, whether the cow had singleton or twins, and the random effect of block. In all analyses, pre-planned single degrees of freedom orthogonal contrasts evaluated the effect of supplementing choline ion as RPC (0 vs. ¼·6.5 + ¼·12.9 + ¼·19.4 + ¼·25.8 g/day), and polynomial contrasts investigated the linear (¾·6.5 + ¼·12.9 vs. ¼·19.4 + ¾·25.8), quadratic (½·6.5 + ½·25.8 vs. ½·12.9 + ½·19.4), and cubic effects (¼·6.5 + ¾·19.4 vs. ¾·12.9 + ¼·25.8) of increasing intake of supplemental choline ion from 6.45 to 25.80 g/day. In all mixed-effects models, the Kenward-Roger method was used to approximate the denominator degrees of freedom to compute the *F*-tests. Data are presented as LSM ± SEM or fold change, unless otherwise stated.

Associations between concentrations of hepatic triacylglycerol and abundance of tran-scripts affected by choline or amount of choline on the hepatic tissue sampled on day 14 were

determined. The dCt for each transcript was analyzed. Because abundance increases as the dCt decreases, the dCt value was multiplied by -1 to facilitate depiction of results in figures in which an increase in -dCt value corresponds to an increased transcript abundance and vice versa. Abundance of transcripts was analyzed using linear mixed-effects models with the MIXED procedure of SAS. The statistical models included the fixed effects of the concentration of hepatic triacylglycerol as linear and quadratic covariates, the BCS at the time of enrollment, and whether the cow had singleton or twins, and the random effect of block. Whenever the effect of the quadratic covariate of hepatic triacylglycerol resulted in $P > 0.10$, it was removed from the statistical model. All other covariates were retained in the final models. Figures were created as scatter plots with the -dCt value in the Y-axis and the hepatic triacylglycerol content in the X-axis using the predicted responses for each cow contributing data for the statistical analysis. The predicted values were obtained from the best linear unbiased estimates including fixed and random effects.

Evidence of statistical significance against the null hypothesis was considered at $P \leq 0.05$, and tendency was considered at $0.05 < P \leq 0.10$.

## Results

A total of 76 cows contributed data for hepatic composition and mRNA abundance in this experiment. Details of the effects of treatment on energy metabolism and blood measures are presented in [7]. Results are depicted in sequence of treatments as 0, 6.5, 12.9, 19.4, and 25.8.

As anticipated, the BCS (3.59 vs. 3.61 vs. 3.60 vs. 3.56 vs. 3.61 ± 0.09), BW (746 vs. 728 vs. 714 vs. 720 vs. 745 ± 27 kg), and days pregnant (210 vs. 212 vs. 210 vs. 211 vs. 209 ± 4 d) when cows were enrolled in the experiment did not differ ($P > 0.10$) among treatments. Treatment did not affect ($P = 0.60$) gestation length in cows (275 vs. 275 vs. 277 vs. 274 vs. 273 ± 2 days), thus resulting in no difference ($P = 0.56$) among treatments in days relative to calving when cows were enrolled in the experiment (64.9 vs. 62.2 vs. 66.4 vs. 62.4 vs. 64.2 ± 3.4 days prepartum).

### Hepatic tissue composition during the ad libitum feeding period

Increasing the amount of choline ion from 6.45 to 25.80 g/day tended to increase linearly ($P = 0.06$) the concentrations of hepatic triacylglycerol from 1.84 to 2.39 ± 0.24 μg/μg DNA, but not when triacylglycerol was expressed on the tissue wet-basis (**Table 1**). Feeding RPC increased ($P = 0.04$) the concentration of hepatic glycogen (6.56 vs. 7.25 ± 0.44% wet-basis), which was observed regardless of the amount of supplemental RPC fed (**Table 1**). Treatments did not affect the ratio of hepatic triacylglycerol to glycogen during the ad libitum period.

### Hepatic tissue mRNA abundance during the ad libitum feeding period

The genes affected ($P \leq 0.10$) by treatment during the ad libitum period and the respective fold changes are reported in **Table 2**. The LSM and respective SEM for the dCt of affected genes are presented in **S3 Table**. Heatmaps with the genes affected by feeding RPC ($P \leq 0.10$) or affected ($P \leq 0.10$) by the linear increase in RPC intake during the ad libitum period are depicted in **Figs 1A** and **2A**, respectively.

Feeding RPC increased ($P < 0.001$) the abundance of a transcript involved in gluconeogenesis (*PCK2*) and tended ($P \leq 0.09$) to upregulate transcripts involved in the biosynthesis of phosphatidylcholine and phosphatidylethanolamine (*CEPT1*) and assembly of VLDL particles (*APOE*). On the other hand, feeding RPC tended ($P \leq 0.08$) to downregulate the abundance of genes involved in the activation of fatty acids (*ACSL1*) and de novo hepatic lipogenesis (*SREBF1*).

**Table 1. Effect of supplementing increased amounts of choline ion as rumen-protected choline on hepatic composition.**

| Item (tissue wet-basis) | Treatment[1] | | | | | | P-value[2] | | | |
|---|---|---|---|---|---|---|---|---|---|---|
| | 0 | 6.5 | 12.9 | 19.4 | 25.8 | SEM | Choline | Linear | Quadratic | Cubic |
| Ad libitum period[3] | | | | | | | | | | |
| Triacylglycerol, μg/μg DNA | 2.06 | 1.84 | 2.15 | 2.26 | 2.39 | 0.24 | 0.70 | 0.06 | 0.61 | 0.79 |
| Triacylglycerol, % | 0.721 | 0.646 | 0.728 | 0.773 | 0.772 | 0.072 | 0.93 | 0.13 | 0.50 | 0.99 |
| Glycogen, % | 6.56 | 6.98 | 7.02 | 7.61 | 7.37 | 0.44 | 0.04 | 0.16 | 0.63 | 0.29 |
| Triacylglycerol to glycogen, ratio | 0.114 | 0.096 | 0.108 | 0.105 | 0.108 | 0.014 | 0.44 | 0.44 | 0.67 | 0.68 |
| Feed restricted period[4] | | | | | | | | | | |
| Triacylglycerol, μg/μg DNA | 15.0 | 14.6 | 10.8 | 13.1 | 9.2 | 2.2 | 0.07 | 0.02 | 0.82 | 0.05 |
| Triacylglycerol, % | 6.67 | 5.45 | 4.68 | 5.13 | 3.81 | 0.92 | 0.007 | 0.04 | 0.51 | 0.20 |
| Glycogen, % | 2.64 | 3.76 | 3.19 | 3.15 | 3.22 | 0.51 | 0.07 | 0.25 | 0.35 | 0.78 |
| Triacylglycerol to glycogen, ratio | 3.53 | 1.82 | 1.73 | 1.98 | 1.35 | 0.63 | 0.003 | 0.39 | 0.43 | 0.45 |
| Relative change[5] | | | | | | | | | | |
| Triacylglycerol | 9.66 | 9.86 | 7.69 | 7.78 | 6.14 | 1.13 | 0.03 | < 0.001 | 0.71 | 0.21 |
| Glycogen | 0.380 | 0.507 | 0.451 | 0.391 | 0.423 | 0.054 | 0.21 | 0.10 | 0.32 | 0.63 |

[1] Supplementation of 0, 6.45, 12.90, 19.35 or 25.80 g/day of choline ion as rumen-protected choline (RPC).

[2] Choline = effect of supplementing choline ion as RPC (0 vs. ¼·6.5 + ¼·12.9 + ¼·19.4 + ¼·25.8 g/day); linear = linear effect of supplementing increasing amounts of choline ion as RPC (¾·6.5 + ¼·12.9 vs. ¼·19.4 + ¾·25.8); quadratic = quadratic effect of supplementing increasing amounts of choline ion as RPC (½·6.5 + ½·25.8 vs. ½·12.9 + ½·19.4); and cubic = cubic effect of supplementing increasing amounts of choline ion as RPC (¼·6.5 + ¾·19.4 vs. ¾·12.9 + ¼·25.8).

[3] Cows were fed for ad libitum intake on days 1 to 5 and hepatic tissue collected on the morning of day 5.

[4] Cows were fed-restricted to 30% of the $NE_L$ required for maintenance and pregnancy from days 6 to 14 according to NASEM [19], and hepatic tissue was collected on day 14.

[5] Relative change in hepatic tissue composition between feed restriction and ad libitum periods (feed restriction/ad libitum).

Abundance of a transcript involved in the regeneration of methionine (*BHMT*) responded quadratically ($P = 0.04$) and another transcript involved in choline metabolism (*MTR*) tended ($P = 0.08$) to respond in a quadratic fashion with amount of RPC supplemented because cows fed 12.90 g/day of choline ion had the least abundance of *BHMT* and *MTR*. Abundance of *PCK2* and a transcript associated with the oxidation of long-chain fatty acids (*PPARA*) increased linearly ($P \leq 0.05$) with increasing intake of choline ion. In addition, abundance of a transcript involved in cholesterol efflux (*ABCA1*) increased linearly ($P = 0.04$) with increasing intake of choline ion, whereas another transcript related to cholesterol metabolism (*FDPS*) tended ($P = 0.08$) to respond quadratically to amount of RPC fed because cows receiving 19.35 g/day of choline ion had the greatest abundance. Abundance of transcripts related to ketogenesis (*HMGCS1*), acute phase response (*CRP*), and urea cycle (*OTC*) tended ($P \leq 0.10$) to respond quadratically to amount of choline supplemented because cows fed 12.90 g/day of choline ion had the smallest abundance of those transcripts. Increasing the amount of choline ion from 6.45 to 25.80 g/day affected quadratically ($P \leq 0.04$) the abundance of transcripts related to oxidative stress (*HMOX2*), lipid peroxidation (*ALDH9A1*), and glycerol phosphate pathway (*GPD2*), and cows fed 19.35 g/day of choline ion had the least abundance for *HMOX2*, whereas those fed 12.90 g/day of choline ion had the least abundance of *ALDH9A1* and *GPD2*.

## Hepatic tissue composition during feed restriction

Feeding RPC tended ($P = 0.07$) to reduce the concentrations of triacylglycerol expressed relative to tissue cellular content (15.0 vs. 11.9 ± 2.2 μg/μg DNA) and reduced ($P = 0.007$) the concentration relative to the tissue wet-basis (6.67 vs. 4.77 ± 0.92% wet-basis; **Table 1**). Feeding RPC tended to increase ($P = 0.07$) the concentrations of glycogen (2.64 vs. 3.33 ± 0.51% wet-

**Table 2. Effect of supplementing increased amounts of choline ion as rumen-protected choline on hepatic relative mRNA abundance of transcripts affected by treatment during the ad libitum period.**

| Item[3] | Treatment[1] | | | | | P-value[2] | | | |
|---|---|---|---|---|---|---|---|---|---|
| | 0 | 6.5 | 12.9 | 19.4 | 25.8 | Choline | Linear | Quadratic | Cubic |
| Choline metabolism | | | | | | | | | |
| BHMT | 1.00 | 1.60 | 0.24 | 0.69 | 1.67 | 0.76 | 0.67 | 0.04 | 0.31 |
| CEPT1 | 1.00 | 1.78 | 1.41 | 2.09 | 1.79 | 0.09 | 0.75 | 0.89 | 0.37 |
| MTR | 1.00 | 1.99 | 0.78 | 1.25 | 2.90 | 0.42 | 0.46 | 0.08 | 0.64 |
| Gluconeogenesis | | | | | | | | | |
| PCK2 | 1.00 | 2.30 | 2.65 | 6.27 | 7.46 | < 0.001 | 0.003 | 0.96 | 0.34 |
| Activation of fatty acids | | | | | | | | | |
| ACSL1 | 1.00 | 0.71 | 0.73 | 0.73 | 0.55 | 0.07 | 0.63 | 0.83 | 0.87 |
| Oxidation of fatty acids | | | | | | | | | |
| PPARA | 1.00 | 0.41 | 0.61 | 0.39 | 1.19 | 0.17 | 0.05 | 0.29 | 0.16 |
| Synthesis of phospholipids | | | | | | | | | |
| AGPAT2 | 1.00 | 1.08 | 1.69 | 2.39 | 2.49 | 0.20 | 0.08 | 0.57 | 0.89 |
| Lipoprotein synthesis and assembly | | | | | | | | | |
| APOE | 1.00 | 2.65 | 2.67 | 3.54 | 4.86 | 0.07 | 0.41 | 0.80 | 0.93 |
| Cholesterol efflux | | | | | | | | | |
| ABCA1 | 1.00 | 1.32 | 0.97 | 2.64 | 3.08 | 0.19 | 0.04 | 0.56 | 0.23 |
| FDPS | 1.00 | 0.89 | 1.50 | 1.55 | 1.15 | 0.51 | 0.51 | 0.08 | 0.62 |
| De novo hepatic lipogenesis | | | | | | | | | |
| SREBF1 | 1.00 | 0.03 | 0.02 | 0.01 | 0.02 | 0.08 | 0.63 | 0.58 | 0.68 |
| Ketogenesis | | | | | | | | | |
| HMGCS1 | 1.00 | 1.31 | 0.97 | 1.14 | 1.49 | 0.30 | 0.44 | 0.10 | 0.64 |
| Acute phase response | | | | | | | | | |
| CRP | 1.00 | 1.21 | 0.70 | 0.84 | 0.99 | 0.69 | 0.60 | 0.08 | 0.41 |
| Oxidative stress | | | | | | | | | |
| HMOX2 | 1.00 | 1.79 | 0.14 | 0.11 | 0.49 | 0.35 | 0.30 | 0.03 | 0.87 |
| Lipid peroxidation | | | | | | | | | |
| ALDH9A1 | 1.00 | 2.72 | 0.59 | 0.66 | 1.76 | 0.81 | 0.63 | 0.04 | 0.77 |
| Glycerol phosphate pathway | | | | | | | | | |
| GPD2 | 1.00 | 0.90 | 0.60 | 1.01 | 1.17 | 0.44 | 0.15 | 0.03 | 0.57 |
| Urea cycle | | | | | | | | | |
| OTC | 1.00 | 3.56 | 0.62 | 0.74 | 1.23 | 0.83 | 0.25 | 0.10 | 0.62 |

[1] Supplementation of 0, 6.45, 12.90, 19.35 or 25.80 g/day of choline ion as rumen-protected choline (RPC).

[2] Choline = effect of supplementing choline ion as RPC (0 vs. ¼·6.5 + ¼·12.9 + ¼·19.4 + ¼·25.8 g/day); linear = linear effect of supplementing increasing amounts of choline ion as RPC (¾·6.5 + ¼·12.9 vs. ¼·19.4 + ¾·25.8); quadratic = quadratic effect of supplementing increasing amounts of choline ion as RPC (½·6.5 + ½·25.8 vs. ½·12.9 + ½·19.4); and cubic = cubic effect of supplementing increasing amounts of choline ion as RPC (¼·6.5 + ¾·19.4 vs. ¾·12.9 + ¼·25.8).

[3] Cows were fed for ad libitum intake on days 1 to 5 and hepatic tissue collected on the morning of day 5.

basis) in the hepatic tissue (Table 1), which resulted in a ratio of triacylglycerol to glycogen that was 2.1-fold smaller ($P = 0.003$) for cows fed RPC compared with non-supplemented cows (3.53 vs. 1.72 ± 0.63). Furthermore, increasing the amount of choline ion from 6.45 to 25.80 g/day reduced ($P \leq 0.04$) linearly the concentrations of triacylglycerol relative to the tissue cellular content or relative to the tissue wet-basis (Table 1). Feeding RPC minimized ($P = 0.03$) the relative increase in concentrations of triacylglycerol in hepatic tissue between the ad libitum and the feed restriction periods (9.66 vs. 7.87 ± 1.13 fold). Furthermore, the relative increase in hepatic triacylglycerol was linearly less ($P < 0.001$) as the amount of supplemental

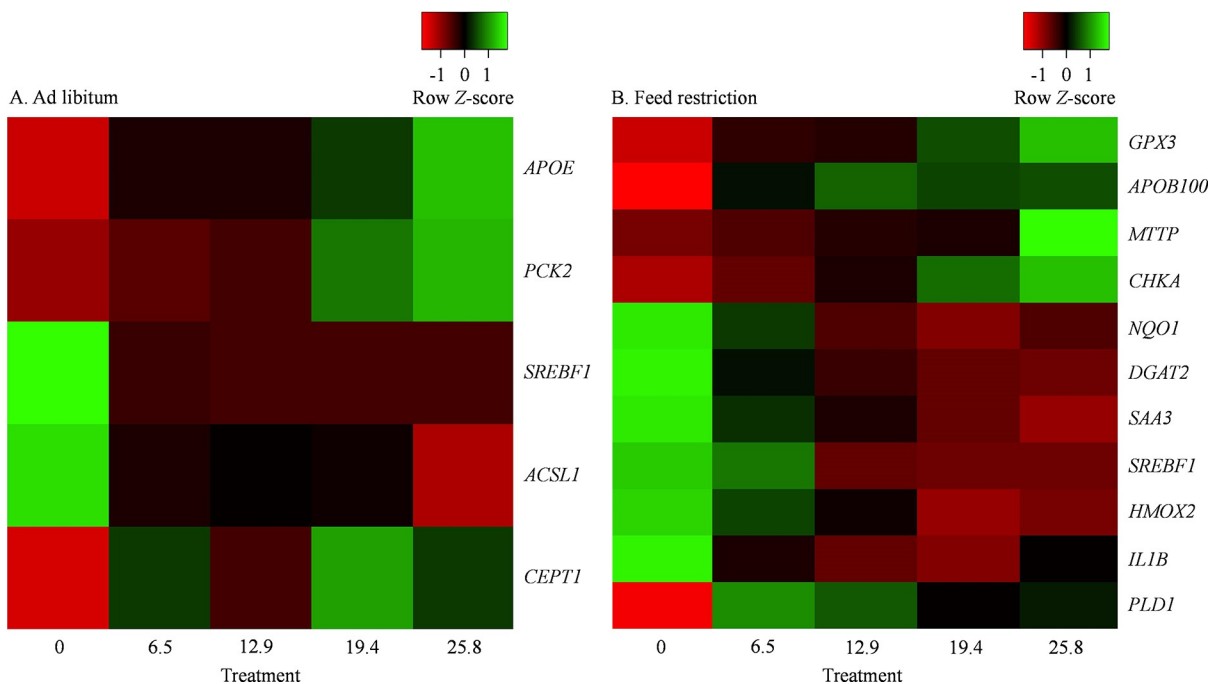

**Fig 1. Heat maps including differently ($P \leq 0.10$) expressed transcripts affected by supplementing choline (0 vs. mean of rest) as rumen-protected choline (RPC).** Cows were supplemented with RPC to supply 0, 6.45, 12.90, 19.35 or 25.80 g/day of choline ion. Panel A represents hepatic tissue sampled on day 5, whereas panel B represents the hepatic tissue sampled on day 14 of the experiment. Transcripts were separated using the average linkage clustering method and following the Euclidean distance measurement approach. The mRNA abundance increases from red to green, and data presented on a Z-score based scaling system.

RPC increased. On the other hand, increasing the amount of choline ion from 6.45 to 25.80 g/day tended ($P = 0.10$) to linearly amplify the decrease in concentrations of glycogen in hepatic tissue between the ad libitum and the feed restriction periods (**Table 1**).

## Hepatic tissue mRNA abundance during feed restriction

The transcripts affected ($P \leq 0.10$) by treatment during the feed restriction period and the respective fold changes are reported in **Table 3**. The LSM and respective SEM for the dCt of affected genes are presented in **S4 Table**. Heatmaps with the transcripts affected ($P \leq 0.10$) by feeding RPC or affected ($P \leq 0.10$) by the linear effect of increasing the amount of RPC fed during the feed restriction period are depicted in **Figs 1B** and **2B**, respectively.

Feeding RPC increased ($P \leq 0.05$) the abundance of transcripts involved in the synthesis of phosphatidylcholine (*CHKA*) and glycerophospholipids (*PLD1*), synthesis and assembly of VLDL particles (*APOB100*), and synthesis of antioxidants (*GPX3*), and tended ($P = 0.08$) to upregulate a transcript involved in the lipidation of VLDL particles (*MTTP*); however, feeding RPC decreased ($P \leq 0.01$) the abundance of transcripts involved in the re-esterification of fatty acids to triacylglycerol (*DGAT2*), de novo hepatic lipogenesis (*SREBF1*), synthesis of cytokines (*IL1B*), and acute phase response (*SAA3*), and tended ($P \leq 0.09$) to downregulate the abundance of a transcript involved in oxidative stress (*HMOX2*) and a transcript related to synthesis of antioxidants (*NQO1*).

Abundance of *CHKA* increased linearly ($P = 0.004$), whereas abundance of *MTR* increased quadratically ($P = 0.04$), and transcripts related to choline metabolism (*CBS*, *PCYT1A*) and fibroblast-growth factor (*FGF21*) tended ($P \leq 0.09$) to respond in a quadratic fashion with increasing intake of choline ion from 6.45 to 25.80 g/day. Abundance of *MTR* and *CBS* transcripts was the greatest when cows consumed 12.90 g/day of choline ion, whereas abundance

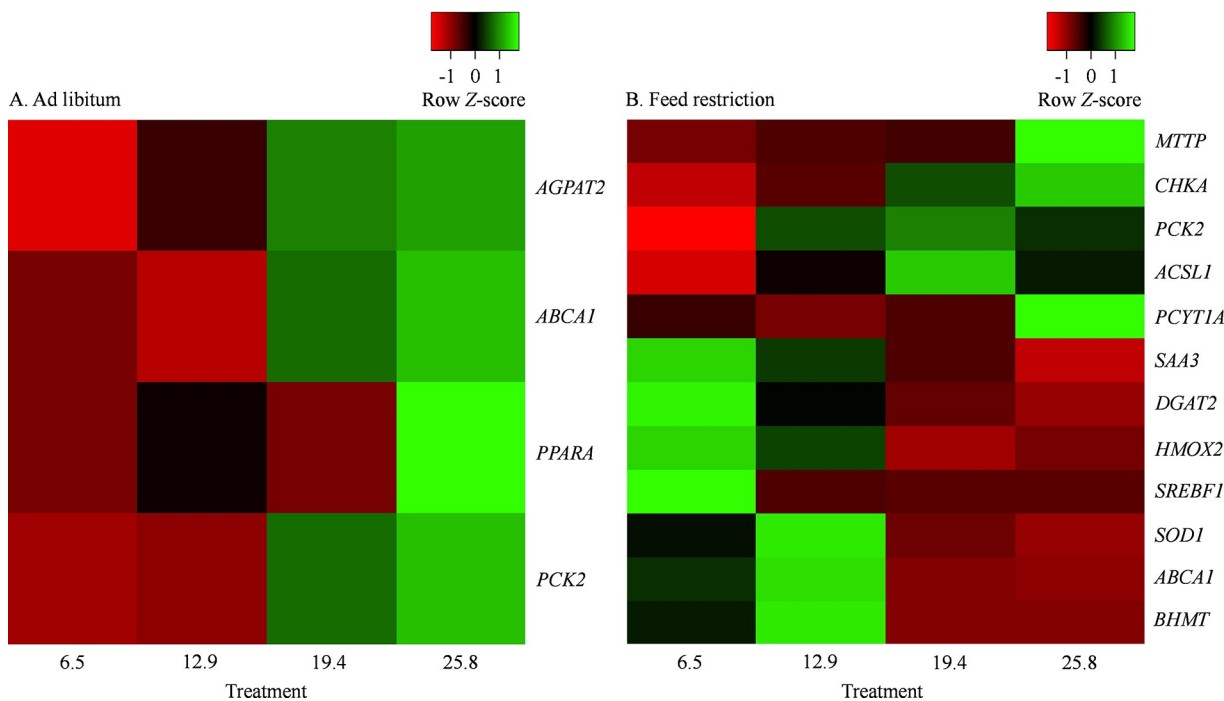

**Fig 2. Heat maps including differently ($P \leq 0.10$) expressed transcripts affected by the linear effect of amount of choline ion supplemented as rumen-protected choline (RPC).** Cows were supplemented with RPC to supply 0, 6.45, 12.90, 19.35 or 25.80 g/day of choline ion and the linear orthogonal contrast evaluated the effect of increasing the amount from 6.45 to 25.80 g/day. Panel A represents hepatic tissue sampled on day 5, whereas panel B represents the hepatic tissue sampled on day 14 of the experiment. Transcripts were separated using the average linkage clustering method and following the Euclidean distance measurement approach. The mRNA abundance increases from red to green, and data presented on a Z-score based scaling system.

of *PCYT1A* and *FGF21* were minimum at the same 12.90 g/day intake of choline ion. Treatment had mixed effects on transcripts involved in gluconeogenesis (*PCK1*, *PCK2*) because increasing intake of choline ion from 6.45 to 25.80 g/day tended ($P \leq 0.10$) to increase *PCK1* quadratically and *PCK2* linearly. In addition, abundance of transcripts related to ketogenesis (*ACAT1*) were affected quadratically ($P = 0.05$), whereas transcripts involved in carnitine metabolism (*CRAT*, *TMLHE*) and oxidation of long-chain fatty acids (*CPT1A*, *PPARA*) tended ($P \leq 0.07$) to be affected quadratically by amount of choline ion consumed, because feeding 12.90 g/day of choline ion maximized the abundance for *ACAT1*, *CPT1A*, and *TMLHE*, whereas feeding 19.35 g/day of choline ion maximized the abundance of *CRAT* and *PPARA*. Abundance of *DGAT2*, *SREBF1*, *SAA3*, and a transcript involved in the synthesis of antioxidants (*SOD1*) decreased linearly ($P \leq 0.05$). Furthermore, abundance of *HMOX2* tended ($P = 0.06$) to decrease in a linear fashion, whereas abundance of *ACSL1* and *MTTP* tended ($P \leq 0.07$) to increase linearly with increasing intake of choline ion from 6.45 to 25.80 g/day. Increasing the amount of choline ion from 6.45 to 25.80 g/day had a cubic ($P = 0.04$) effect on abundance of *BHMT*, and tended ($P \leq 0.09$) to have a cubic effect on abundance of *ABCA1* and *HMGCL3*, the latter a transcript involved in ketogenesis.

## Associations between hepatic triacylglycerol and mRNA abundance during feed restriction

Linear associations ($P \leq 0.05$) were observed between the concentration of hepatic triacylglycerol and abundance of *CHKA*, *PCYT1A*, *APOB100*, *MTTP* and *SAA3* (**Fig 3A–3E**). As the

**Table 3. Effect of supplementing increased amounts of choline ion as rumen-protected choline on hepatic relative mRNA abundance of transcripts affected by treatment during the feed restriction period.**

| Item[3] | Treatment[1] | | | | | P-value[2] | | | |
|---|---|---|---|---|---|---|---|---|---|
| | 0 | 6.5 | 12.9 | 19.4 | 25.8 | Choline | Linear | Quadratic | Cubic |
| Choline metabolism | | | | | | | | | |
| BHMT | 1.00 | 2.64 | 5.27 | 0.80 | 0.68 | 0.41 | 0.005 | 0.36 | 0.04 |
| CBS | 1.00 | 0.46 | 5.73 | 1.67 | 0.75 | 0.81 | 0.96 | 0.08 | 0.27 |
| CHKA | 1.00 | 2.82 | 4.80 | 8.05 | 10.27 | < 0.001 | 0.004 | 0.68 | 0.87 |
| MTR | 1.00 | 0.33 | 1.44 | 1.23 | 0.51 | 0.62 | 0.64 | 0.04 | 0.73 |
| PCYT1A | 1.00 | 1.42 | 1.12 | 1.34 | 2.95 | 0.18 | 0.07 | 0.09 | 0.87 |
| PLD1 | 1.00 | 2.41 | 2.25 | 1.89 | 1.99 | 0.05 | 0.60 | 0.86 | 0.82 |
| Growth factor | | | | | | | | | |
| FGF21 | 1.00 | 0.89 | 0.56 | 0.56 | 0.84 | 0.18 | 0.85 | 0.07 | 0.97 |
| Gluconeogenesis | | | | | | | | | |
| PCK1 | 1.00 | 1.24 | 1.98 | 1.90 | 0.93 | 0.32 | 0.51 | 0.07 | 0.91 |
| PCK2 | 1.00 | 0.76 | 1.37 | 1.45 | 1.30 | 0.52 | 0.10 | 0.13 | 0.71 |
| Carnitine metabolism | | | | | | | | | |
| CRAT | 1.00 | 0.55 | 1.82 | 2.23 | 0.35 | 0.95 | 0.76 | 0.07 | 0.77 |
| TMLHE | 1.00 | 0.36 | 1.19 | 0.92 | 0.49 | 0.42 | 0.75 | 0.06 | 0.60 |
| Activation of fatty acids | | | | | | | | | |
| ACSL1 | 1.00 | 0.76 | 1.16 | 1.61 | 1.24 | 0.59 | 0.07 | 0.13 | 0.62 |
| Oxidation of fatty acids | | | | | | | | | |
| CPT1A | 1.00 | 0.28 | 4.10 | 2.07 | 1.28 | 0.78 | 0.27 | 0.06 | 0.37 |
| PPARA | 1.00 | 0.61 | 2.74 | 3.21 | 1.16 | 0.51 | 0.47 | 0.06 | 0.95 |
| Re-esterification of fatty acids | | | | | | | | | |
| DGAT2 | 1.00 | 0.39 | 0.18 | 0.08 | 0.04 | < 0.001 | < 0.001 | 0.94 | 0.91 |
| Lipoprotein synthesis and assembly | | | | | | | | | |
| APOB100 | 1.00 | 4.78 | 5.86 | 5.38 | 5.62 | < 0.001 | 0.81 | 0.83 | 0.81 |
| MTTP | 1.00 | 1.73 | 2.39 | 2.58 | 7.91 | 0.08 | 0.06 | 0.43 | 0.57 |
| Cholesterol efflux | | | | | | | | | |
| ABCA1 | 1.00 | 1.90 | 3.57 | 0.40 | 0.22 | 0.83 | < 0.001 | 0.24 | 0.07 |
| De novo hepatic lipogenesis | | | | | | | | | |
| SREBF1 | 1.00 | 0.74 | 0.04 | 0.03 | 0.02 | < 0.001 | 0.001 | 0.50 | 0.33 |
| Ketogenesis | | | | | | | | | |
| ACAT1 | 1.00 | 1.05 | 1.70 | 1.65 | 1.15 | 0.20 | 0.78 | 0.05 | 0.86 |
| HMGCL3 | 1.00 | 0.84 | 1.10 | 0.60 | 0.95 | 0.59 | 0.83 | 0.71 | 0.09 |
| Cytokines | | | | | | | | | |
| IL1B | 1.00 | 0.64 | 0.54 | 0.50 | 0.67 | 0.01 | 0.33 | 0.37 | 0.84 |
| Acute phase response | | | | | | | | | |
| SAA3 | 1.00 | 0.52 | 0.33 | 0.16 | 0.03 | 0.001 | < 0.001 | 0.24 | 0.81 |
| Oxidative stress | | | | | | | | | |
| HMOX2 | 1.00 | 0.67 | 0.47 | 0.17 | 0.22 | 0.09 | 0.06 | 0.57 | 0.43 |
| Synthesis of antioxidants | | | | | | | | | |
| GPX3 | 1.00 | 3.52 | 3.74 | 5.57 | 7.40 | 0.01 | 0.29 | 0.84 | 0.85 |
| NQO1 | 1.00 | 0.53 | 0.18 | 0.05 | 0.17 | 0.06 | 0.23 | 0.16 | 0.46 |

(*Continued*)

**Table 3.** (Continued)

| Item[3] | Treatment[1] | | | | | P-value[2] | | | |
|---|---|---|---|---|---|---|---|---|---|
| | 0 | 6.5 | 12.9 | 19.4 | 25.8 | Choline | Linear | Quadratic | Cubic |
| SOD1 | 1.00 | 1.45 | 2.53 | 0.88 | 0.67 | 0.66 | 0.05 | 0.29 | 0.18 |

[1] Supplementation of 0, 6.45, 12.90, 19.35 or 25.80 g/day of choline ion as rumen-protected choline (RPC).

[2] Choline = effect of supplementing choline ion as RPC (0 vs. ¼·6.5 + ¼·12.9 + ¼·19.4 + ¼·25.8 g/day); linear = linear effect of supplementing increasing amounts of choline ion as RPC (¾·6.5 + ¼·12.9 vs. ¼·19.4 + ¾·25.8); quadratic = quadratic effect of supplementing increasing amounts of choline ion as RPC (½·6.5 + ½·25.8 vs. ½·12.9 + ½·19.4); and cubic = cubic effect of supplementing increasing amounts of choline ion as RPC (¼·6.5 + ¾·19.4 vs. ¾·12.9 + ¼·25.8).

[3] Cows were fed-restricted to 30% of the $NE_L$ required for maintenance and pregnancy from days 6 to 14 according to NASEM [19], and hepatic tissue was collected on day 14.

concentration of hepatic triacylglycerol increased, the abundance of *CHKA*, *PCYT1A*, *APOB100*, and *MTTP* decreased linearly ($P \leq 0.02$). On the other hand, as the concentration of hepatic triacylglycerol increased, the abundance of *SAA3* increased linearly ($P = 0.05$). Hepatic triacylglycerol tended to be associated quadratically ($P = 0.06$) with abundance of *DGAT2* (**Fig 3F**). As the concentration of triacylglycerol in hepatic tissue increased, the abundance of *DGAT2* increased and plateaued at concentrations ranging between 8 and 10%, after which abundance began to decline (**Fig 3F**).

A schematic diagram of the effects of choline on hepatic tissue transcript abundance during negative nutrient balance is depicted in **Fig 4**.

## Discussion

Feed restriction altered hepatic composition such that cows accumulated triacylglycerol and mobilized glycogen; however, supplementing choline ion as RPC attenuated the increase in triacylglycerol and increased glycogen content in the hepatic tissue. The observed changes in hepatic composition were linked with changes in key transcripts involved in metabolism of choline, synthesis and assembly of VLDL particles, hepatic lipogenesis, oxidative stress, and inflammation. Feeding RPC upregulated *CHKA*, which plays a role in phosphorylation of choline for synthesis of phosphocholine, the first step to provide substrate for phospholipid synthesis from absorbed dietary choline. Also, increasing the amount of RPC fed tended to increase *PCYT1A*, an enzyme that catalyzes the rate-limiting reaction in the Kennedy pathway by transferring cytidine triphosphate to phosphocholine to synthesize cytidine diphosphate choline or CDP-choline [26]. It is possible that the hepatic tissue of cows supplemented with RPC had increased supply of choline ion to be processed by the enzymatic machinery of the liver for de novo synthesis of phosphatidylcholine during the period of extensive lipomobilization. Feeding RPC or abomasally infusing choline chloride increased the net portal flux of choline ions [27], and feeding RPC increased the concentrations of choline biomolecules in plasma of cows [7]. Increasing de novo synthesis of phosphatidylcholine might benefit lipid transport within cells and favor the synthesis and assembly of lipoproteins for export of triacylglycerol from the hepatic tissue, which is expected to reduce the degree of hepatic lipidosis as shown in the schematic **Fig 4**.

The liver of ruminants is known to have limited rate of secretion of VLDL [28], which might be part of the reason for the increased risk of dairy cows developing hepatic steatosis during periods of extensive lipomobilization. Assembly and secretion of VLDL by hepatocytes require transfer of triacylglycerols to the lipoprotein, incorporation of phospholipids such as phosphatidylcholine, and presence of apolipoproteins such as APOB100. Both MTTP and APOB100 proteins are essentially required for the lipidation and stabilization of newly

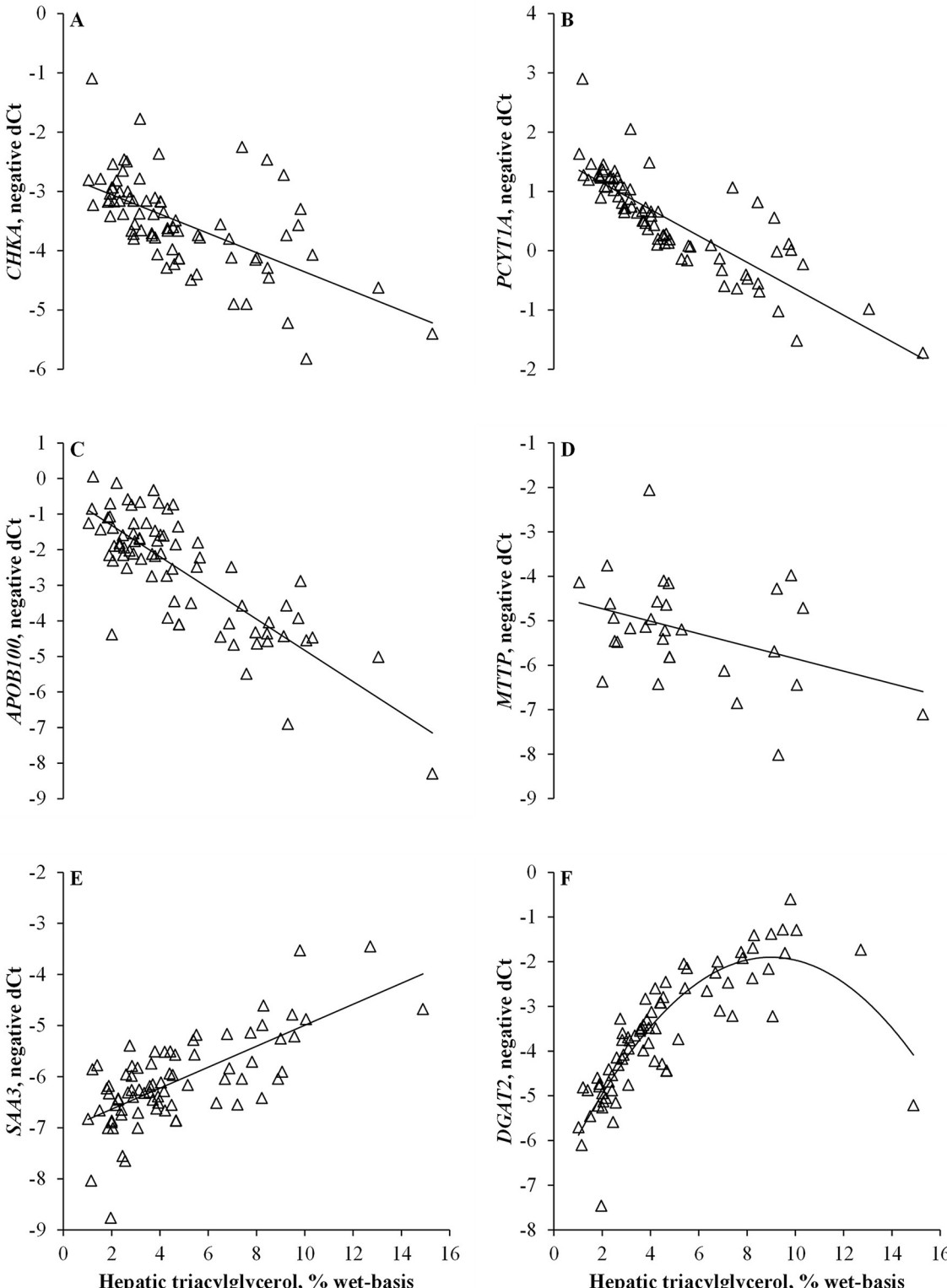

**Fig 3.** Associations between concentrations of hepatic triacylglycerol and mRNA abundance based on the negative delta cycle threshold (-dCt) for *CHKA* (A), *PCYT1A* (B), *APOB100* (C), *MTTP* (D), *SAA3* (E), and *DGAT2* (F) on day 14 of the experiment. Panel A: Linear association ($P = 0.02$); panel B: Linear association ($P = 0.001$); panel C: Linear association ($P < 0.001$); panel D: Linear association ($P = 0.02$); panel E: Linear association ($P = 0.05$); and panel F: Linear ($P = 0.009$) and quadratic ($P = 0.06$) associations.

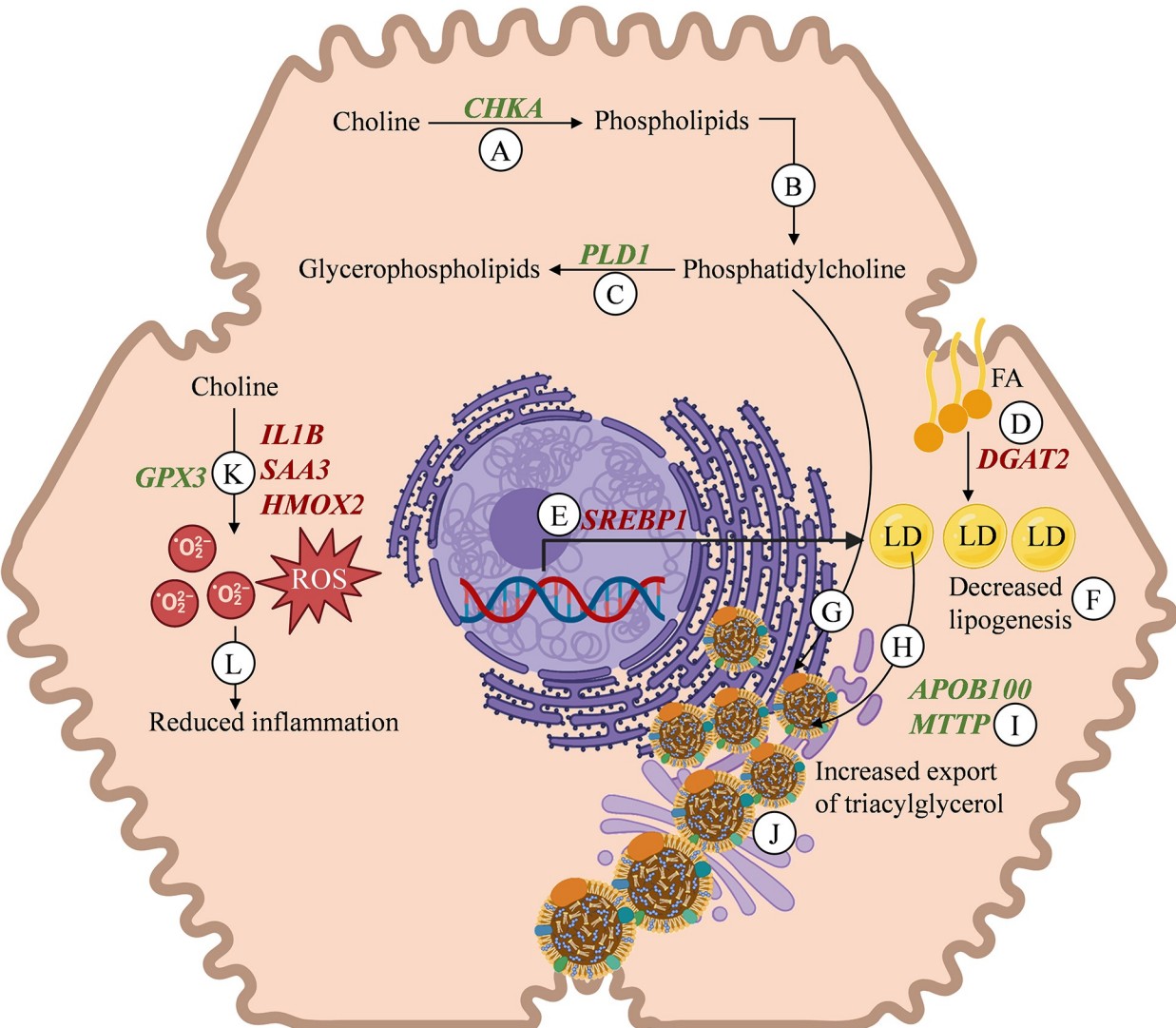

**Fig 4. Schematic representation of effect of choline as rumen-protected choline (RPC) on the mRNA abundance of transcripts in the hepatocyte during negative nutrient balance.** Feeding choline increased the abundance of *CHKA* (**A**), which should favor an increased synthesis of phospholipids such as phosphatidylcholine (**B**). Feeding choline increased the mRNA abundance of *PLD1*, which is involved in the hydrolysis of phosphatidylcholine to produce glycerophospholipids (**C**). Supplementing choline reduced the transcripts abundance of *DGAT2* (**D**) and *SREBP1* (**E**), which indicates less machinery for re-esterification of fatty acids (**FA**) to triacylglycerols, thus reducing the formation of lipid droplets (**LD**) in the hepatocyte (**F**). An increased availability of phosphatidylcholine might be used for the synthesis of cell or organelle membranes or to be a component of lipoproteins (**G**). The LD might be incorporated preferentially in the core of very-low density lipoprotein (**VLDL**) particles (**H**). The increased abundance of *APOB100* and *MTTP* (**I**) in choline fed cows is expected to favor the synthesis and assembly of VLDL that eventually increases the export of triacylglycerol (**J**). Feeding choline reduced the abundance of *IL1B*, *SAA3*, and *HMOX2*, whereas it increased the abundance of *GPX3* (**K**), suggesting reduced inflammatory signals involved in the acute phase response, concurrent with enhanced antioxidant protection to scavenge reactive oxygen species (**ROS**) and limit the injury to hepatocytes (**L**). Abbreviations: *APOB100* = apolipoprotein B100; *CHKA* = choline kinase alpha; *DGAT2* = diacylglycerol O-acyltransferase 2; *GPX3* = glutathione peroxidase 3; *HMOX2* = heme oxygenase 2; *IL1B* = interleukin 1 beta; *MTTP* = microsomal triglyceride transfer protein; *PLD1* = phospholipase D1; *SAA3* = serum amyloid A3; *SREBP1* = sterol regulatory element binding protein 1. The schematic diagram was created using BioRender (https://www.biorender.com/).

synthesized VLDL particles [29]. The abundance of *APOB100* and *MTTP* were upregulated by supplementing RPC. Also, during feed restriction, increased concentrations of hepatic triacyl-glycerol was linearly associated with reduced abundance of *APOB100* and *MTTP*. Such response possibly suggests that either cows with reduced abundance of those transcripts are

more prone to hepatic steatosis, or that hepatic steatosis suppresses abundance of *APOB100* and *MTTP* further compromising the ability of hepatocytes to export triacylglycerols during periods of risk of hepatic steatosis. Dairy cows with fatty liver had reduced mRNA and protein abundance of APOB100, APOE, and MTTP [30]. When primary bovine hepatocytes were treated in vitro with increasing concentrations of choline chloride, from 0 to 4,528 μ*M*, the authors observed an increased secretion of VLDL in the medium [31], which is expected to minimize the risk of fatty acid re-esterification and deposition of triacylglycerol as fat droplets. Herein, increasing the intake of choline ion from 6.45 to 25.80 g/day linearly reduced the content of hepatic triacylglycerol, likely because of increased export as lipoproteins [31,32]. It is interesting that the changes in hepatic triacylglycerol content with supplemental RPC were observed independent of changes in plasma concentrations of fatty acids or β-hydroxybutyrate [7]. These findings corroborate those of Arshad et al. [33], who also observed that a diet supplemented with RPC attenuated the degree of hepatic lipidosis independent of the degree of lipomobilization in cows subjected to feed restriction. It is possible that feeding RPC supplies substrate for phospholipids synthesis and stimulates hepatic abundance of key regulatory transcripts and proteins needed for hepatic lipoprotein secretion, thus promoting hepatic lipotropic effects in dairy cows.

Both feeding RPC and increasing the amount of RPC markedly reduced the abundance of *DGAT2* and *SREBF1* during feed restriction, thus suggesting reduced machinery for de novo hepatic lipogenesis. Indeed, the quadratic association between hepatic triacylglycerol and abundance of *DGAT2* suggests that this enzyme might play a role in re-esterification of fatty acids and triacylglycerol deposition into the hepatic tissue. The biosynthesis of triacylglycerol requires precursors such as fatty acyl-CoAs, which are acylated with the two free hydroxyl groups of L-glycerol 3-phosphate to yield diacylglycerol 3-phosphate or phosphatidic acid. Phosphatidic acid can either undergo hydrolysis by phosphatidic acid phosphatase to form 1,2-diacylglycerol, which is converted to triacylglycerols, or combine with the head groups such as choline, serine, or ethanolamine to synthesize glycerophospholipids. Because RPC reduced abundance of *DGAT2* and *SREBF1*, but increased that of *CHKA* and *PLD1*, it is likely that choline reduced the re-esterification of fatty acids to triacylglycerol and favored the increased synthesis of choline-derived phospholipids (**Fig 4**). An increase in the biosynthesis of phospholipids, such as phosphatidylcholine not only facilitates the export of triacylglycerol from hepatic tissue, but also possesses anti-inflammatory properties [17].

Supplementing RPC reduced the abundance of *IL1B* and *SAA3* suggesting a reduced acute phase response compared to non-supplemented cows. Indeed, supplementing RPC reduced the concentrations of haptoglobin in plasma [7,33]. A linear positive association was observed between concentrations of hepatic triacylglycerol and mRNA abundance of *SAA3*, which reinforces the fact that excess lipidosis might trigger a proinflammatory response during periods of lipomobilization. Feeding 12.90 g/day of choline ion during the transition period decreased the blood concentrations of pro-inflammatory cytokines [34], increased the phagocytic capacity in monocytes [35], and decreased the activation of in vitro stimulated immune cells in the early postpartum period [36]. Feeding choline to rats attenuated the TNFA response to an intravenous injection of lipopolysaccharides [37]. The authors showed that supplementing choline precluded an increase in concentrations of aspartate aminotransferase and alanine transaminase in serum after a lipopolysaccharide challenge, which suggest reduced hepatocyte lysis [37]. Zenobi et al. [38] showed that calves born from dams supplemented with RPC during the last 3 weeks of gestation had lesser inflammatory response to a lipopolysaccharide challenge than calves born from dams not supplemented with RPC. These findings suggest that feeding choline ion as RPC might attenuate the degree of inflammatory response, which might favor hepatic function and minimize the risk of fatty liver in dairy cows. It is well established

that pro-inflammatory mediators increase hepatic triacylglycerol independent of their effects on appetite of cows [16].

Feeding RPC linearly reduced the abundance of *HMOX2*, a transcript which participates in the synthesis of the enzyme heme oxygenase-2 that is important for acute inflammatory responses [39]. The reduced abundance of *HMOX2* might attenuate oxidative damage to hepatocytes and provide hepatoprotective effects. On the other hand, supplementation of RPC increased the abundance of *GPX3* indicating increased mRNA needed for the synthesis of antioxidants that might facilitate the scavenging of free radicals and improve the cellular antioxidant protection during periods of extensive lipomobilization in cows.

As the intake of choline ion increased from 6.45 to 25.80 g/day during feed restriction, the abundance of *ACSL1* increased. Long-chain fatty acyl-CoA synthetase 1 transcript is involved in the activation of fatty acids that might increase the availability of fatty acyl-CoAs for synthesis of acylcarnitines, which are obligatory cofactors enabling the transport of long-chain fatty acids across the inner mitochondrial membrane for oxidation. Feeding increasing amounts of carnitine during the transition period to multiparous cows increased the concentrations of carnitine in the hepatic tissue and reduced the accumulation of triacylglycerol in hepatic tissue, primarily through a stimulation of oxidation of long-chain fatty acids within the liver slices compared to non-supplemented cows [40]. In the present experiment, increasing the amount of choline ion from 6.45 to 25.80 g/day increased the abundance of *CRAT* and *TMLHE*, which code for proteins required for the biosynthesis of carnitine needed for the transport of fatty acyl-CoAs to the mitochondrion in hepatic tissue. Increasing the amount of choline ion from 6.45 to 25.80 g/day increased the abundance of *CPT1A* and *PPARA* in a quadratic fashion, which might favor β-oxidation of those fatty acids. A possible increase in the availability of carnitine, concurrent with enhanced message for β-oxidation of long-chain fatty acids, might reduce the re-esterification of fatty acids to triacylglycerol and minimize the severity of hepatic lipidosis in RPC fed cows.

Feeding RPC linearly increased the abundance of *PCK1* during the ad libitum period, whereas abundance of *PCK1* and *PCK2* increased during feed restriction, suggesting increased availability of enzymes for gluconeogenesis. It is interesting that supplementation of RPC increased the glycogen contents in the liver during ad libitum and feed restriction periods. It is possible that choline effects on phosphoenolpyruvate carboxykinase favored gluconeogenesis that spared glucose to be partitioned to glycogenesis, or decreased glycogenolysis. Increased glycogen content in the hepatic tissue of cows fed RPC has been previously observed [33,41]. Supplementation of RPC resulted in a quadratic effect on abundance of *CBS* and *MTR*, which code for proteins required for the regeneration of methionine. Increased regeneration of methionine leads to the formation of S-adenosyl methionine, which acts as a universal methyl donor and might supply methyl groups for synthesis of phosphatidylcholine through the phosphatidylethanolamine N-methyltransferase pathway or alter methylation of proteins and DNA that can affect function of proteins or abundance of transcripts. Cai et al. [42] fed betaine to sows throughout gestation and showed that the hepatic tissue of piglets born to betaine supplemented sows had increased glycogen content and abundance of genes involved in gluconeogenesis compared with piglets born to non-supplemented sows. Chandler and White [43] isolated primary hepatocytes from neonatal calves and treated them with increasing concentration of choline chloride. The authors showed that hepatic content of glycogen increased with increased supplementation of choline in the culture medium. It is suggested that methyl groups derived from choline, after oxidation to betaine, are directly tied to the metabolism of glucogenic amino acids such as glycine and serine and such glucogenic amino acids might favor the cellular glycogenesis, which might contribute for the maintenance of hepatic glycogen concentrations in dairy cows.

## Conclusions

Supplementing the diet of dairy cows with choline ion as RPC attenuated the degree of hepatic steatosis during feed restriction. Furthermore, supplementing RPC altered the abundance of several transcripts in the hepatic tissue, in particular increased the amount of message for transcripts involved in lipotropic effects. The changes in hepatic composition were associated with changes in abundance of several transcripts linked with phosphatidylcholine synthesis and VLDL assembly and secretion, which might be linked to the beneficial effects of feeding increased amounts of choline ion as RPC to reduce hepatic steatosis. In addition to a potential increase in the export of lipids by feeding choline, increasing the amount ingested by cows increased the abundance for transcripts involved in the synthesis of carnitine and β-oxidation of fatty acids, which might play a role in reducing re-esterification to triacylglycerol, thus minimizing the severity of fatty liver. The increased abundance of transcripts involved in the synthesis of phosphatidylcholine and antioxidants coincided with the reduced abundance of transcripts associated with acute phase response or oxidative damage, thus providing evidence to hepatocellular protection by choline in dairy cows. One of the observed effects of supplementing RPC was increased abundance of transcripts related to gluconeogenesis, which might explain the increments in glycogen content in the liver perhaps by altering the fate of carbon to favor glycogenesis. Collectively, feeding 25.80 g/d of choline ion as RPC during negative nutrient balance promotes more pronounced lipotropic effects that minimizes the risk of fatty liver, and such effects were linked with altered transcript abundance for pathways related to transport, export, or oxidation of lipids in hepatocytes.

## Supporting information

**S1 Table. Dietary ingredients and nutrient content of diets fed during the ad libitum and feed restriction periods.**
(DOCX)

**S2 Table. List of transcripts and primers investigated by real time PCR analysis of hepatic tissue segregated according to function.**
(DOCX)

**S3 Table. Effect of amount of choline ion supplemented as rumen-protected choline on delta cycle threshold values of transcripts affected by treatment during the ad libitum period (LSM and SEM).**
(DOCX)

**S4 Table. Effect of amount of choline ion supplemented as rumen-protected choline on delta cycle threshold values of transcripts affected by treatment during the feed restriction period (LSM and SEM).**
(DOCX)

## Acknowledgments

The authors thank Jorge E. Zuniga and Michael B. Poindexter (University of Florida) for their help ensuring proper daily care of cows and sample collection. The help of the staff of the University of Florida Dairy Unit (Hague, FL) is greatly acknowledged. The authors thank Dr. Barbara Barton of Balchem Animal Nutrition and Health (New Hampton, NY) for providing the rumen-protected choline product for this experiment and Dr. Peter J. Hansen (University of Florida) for the use of his laboratory for the extraction of RNA.

## Author Contributions

**Conceptualization:** Usman Arshad, Marcos G. Zenobi, Charles R. Staples, José E. P. Santos.

**Data curation:** Usman Arshad, Marcos G. Zenobi.

**Formal analysis:** Usman Arshad, José E. P. Santos.

**Funding acquisition:** Charles R. Staples, José E. P. Santos.

**Investigation:** Marcos G. Zenobi, Paula Tribulo, Charles R. Staples, José E. P. Santos.

**Methodology:** Paula Tribulo, Charles R. Staples, José E. P. Santos.

**Project administration:** Usman Arshad, Charles R. Staples.

**Resources:** Charles R. Staples, José E. P. Santos.

**Supervision:** José E. P. Santos.

**Writing – original draft:** Usman Arshad, José E. P. Santos.

**Writing – review & editing:** Usman Arshad, Marcos G. Zenobi, Paula Tribulo, José E. P. Santos.

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
