## [Decision Letter · Decision Letter 0]

23 May 2023

PONE-D-23-11577Effect of amount of rumen-protected choline on hepatic metabolism during induction of fatty liver in dairy cowsPLOS ONE

Dear Dr. Santos,

Thank you for submitting your manuscript to PLOS ONE. After careful consideration, we feel that it has merit but does not fully meet PLOS ONE’s publication criteria as it currently stands. Therefore, we invite you to submit a revised version of the manuscript that addresses the points raised during the review process.

 The article is well-written and attractive in this field. Before acception, please address all the questions raised by Reviewer.

We look forward to receiving your revised manuscript.

Kind regards,

Fei Luo

Academic Editor

PLOS ONE

Journal Requirements:

3. To comply with PLOS ONE submissions requirements, in your Methods section, please provide additional information regarding the experiments involving animals and ensure you have included details on (1) methods of sacrifice, (2) methods of anesthesia and/or analgesia, and (3) efforts to alleviate suffering.

"The authors thank Jorge E. Zuniga and Michael B. Poindexter (University of Florida) for their help ensuring proper daily care of cows and sample collection. The help of the staff of the University of Florida Dairy Unit (Hague, FL) is greatly acknowledged. The authors thank Barbara Barton of Balchem Animal Nutrition and Health (New Hampton, NY) for providing the rumen-protected choline product for this experiment and Dr. Peter J. Hansen (University of Florida) for allowing the use of his laboratory for the extraction of RNA. The Higher Education Commission of Pakistan partially funded the Ph.D. program of Usman Arshad. The authors have not stated any conflicts of interest."

"The Higher Education Commission of Pakistan partially funded the Ph.D. program of Usman Arshad."

6. Thank you for stating the following financial disclosure: 

"The Higher Education Commission of Pakistan partially funded the Ph.D. program of Usman Arshad."

7. We note that you have stated that you will provide repository information for your data at acceptance. Should your manuscript be accepted for publication, we will hold it until you provide the relevant accession numbers or DOIs necessary to access your data. If you wish to make changes to your Data Availability statement, please describe these changes in your cover letter and we will update your Data Availability statement to reflect the information you provide."

Reviewers' comments:

Reviewer's Responses to Questions

**Comments to the Author**

1. Is the manuscript technically sound, and do the data support the conclusions?

Reviewer #1: Yes

Reviewer #2: Yes

2. Has the statistical analysis been performed appropriately and rigorously? 

Reviewer #1: Yes

Reviewer #2: Yes

3. Have the authors made all data underlying the findings in their manuscript fully available?

Reviewer #1: Yes

Reviewer #2: Yes

4. Is the manuscript presented in an intelligible fashion and written in standard English?

Reviewer #1: Yes

Reviewer #2: Yes

5. Review Comments to the Author

Reviewer #1: The contents of the present manuscript add on to the current knowledge of supplementing choline to dairy cows. The study is well performed and of relevance and interest not only for dairy science. However, there are some aspects as detailed below that should be addressed to improve the manuscript.

Title: admittedly, I am not a native speaker, but the term “amount” in the title sounds odd to me. To my understanding, four different doses of choline were tested, so something like “Dose-dependent effects of rumen-protected choline on hepatic 1 metabolism during induction of fatty liver in dairy cows” would be more adequate.

Abstract.

Line 20: is it important to stress Choline ION here? Why? In the later text, it was stated that choline chloride was used. Though it will likely dissociate in watery solutions, the ion was not fed as such to the animals and thus the respective statement in the abstract is not exact.

What kind of rumen protection was used? Was the carrier feed accounted for its energy content, i.e., by administration of the carrier feed without the active compound to have the same amount of top-dressing in all four groups?

Line 22: Were primiparous or pluriparous cows enrolled, or both? Information on this is provided later, but should also be included in the abstract.

Line 23: the digits in the dose information should be limited to three, more is not more precise (e.g., 12.9 instead of 12.90). Use numbers as you did in bold print in Lines 110 – 112.

Line 24; It is not quite clear to what the day mentioned refer, is it day of lactation? In any case, please indicated accordingly. How were the cows fed before calving?

Line 27: “mRNA expression” cannot by directly measured, only mRNA abundance. This should be stated as such and also some specification of the target genes should be provided.

Line 32: what means “as is basis”?

Line 33: change expression to abundance and do so throughout the entire manuscript.

Line 42: last week or weeks??

Line 53 – 55: is supplementing choline as inferred in these lines always physiologically necessary for ruminants, or only for high yielding dairy cows?

Line 71: see comment to the title (amounts)

Line 75: delete “of” after dispose

Line 97 – 99. The diet of the pregnant dry cows is not mentioned; assumingly they were not fed ad libitum and if the diet was changed only with calving, this would affect the adaptive capability of the animals. At least the energy content of the dry cows` diet should be provided.

Lines 103 – 104: the days of the experiment seem to equal with days of lactation. However, this would imply that the choline supplementation also started only at day 1 immediately after calving. Please make it very clear what day 1 of the experiment is (see also Lines 117 – 118).

Line 146 vs. Line 149: the days of sampling are not consistent.

Line 150: 25 mg were (not was)

Lines 158 – 161: what about RNA quality, e.g., RIN numbers?

Line 354: change to phospholipid (singular form)

Lines 353 – 357: Provide adequate reference (-s) for the statements about the Kennedy pathway, not every potential reader would know it.

Line 373: change “expression” to “mRNA abundance” (see also Line 376, and several cases in the following sections, and also the figure captions)

Line 381: could you relate the micro molar concentrations from this in vitro study to the blood (and maybe liver) concentrations in the cows from the present study? Would they be in the same order of magnitude?

Lines 424 – 426: would attenuating the degree of inflammatory response by RPC be always beneficial, could you speculate on reduced immune response in case of infections?

Line 441: what are “parous cows”?

Line 469: in this case choline ion?? Hepatic is rather hepatocytic?

Conclusions: coming back to the title, could you conclude which dosage is optimal in your feed restriction challenge, and may be without such challenge?

References: ref. No 4 is problematic since only under review, so you can`t use it as such. Any progress with that one?

Figure 4: which software was used to generate the schematic? In case of Biorender it would need to be referenced. Moreover, in this visualization, it would be helpful to have the full gene names without having to jump to the supplementary information.

Reviewer #2: The article is a very nice contribution to the study of dosage of RPC to cows subjected to fatty liver and to the molecular action of choline.

Abstract:

L21: “…subjected to a fatty liver induction protocol.”

L21: Define: Pregnant, non-lactating Holstein cows with at least one lactation.

L23-24: Define the duration of RPC supplementation (14 d ? 5 ad lib + 9 feed restricted) and how it was supplemented to guarantee the intake of the daily dosages (Top-dress).

Material and Methods:

L111: Describe product ReaShure: company, country.

L113-115: Make clear that the treatments were fed only for 14 days (5 + 9).

L124: Is it NASEM 2021 predicted supply ? I would report the predicted metabolizable AA efficiency, if this model was used to report Met supply.

L125: Avoid paragraph with only one sentence.

L135: I would report the actual DMI ± SD during the period of feed restriction (kg/d and % of BW).

L137: I would report metabolizable MET predicted efficiency in addition to the prediction of g/d.

Results:

L261: P < 0.01. No need to use 3 decimals (0.001).

L285: P < 0.01. In place of P = 0.007.

L289: P < 0.01.

L334: Avoid paragraph with only one sentence.

Discussion:

Is it possible to provide some discussion on why RPC increased (P = 0.06 for linear) triacylglycerol (ug/ug DNA) during ad libitum feeding (L249-250) and reduced it during feed restriction (L284-285) ? There was a cubic reduction during feed restriction (P = 0.05). Does it have a biological meaning ?

6. PLOS authors have the option to publish the peer review history of their article (what does this mean?). If published, this will include your full peer review and any attached files.

Reviewer #1: No

Reviewer #2: No

While revising your submission, please upload your figure files to the Preflight Analysis and Conversion Engine (PACE) digital diagnostic tool, https://pacev2.apexcovantage.com/. PACE helps ensure that figures meet PLOS requirements. To use PACE, you must first register as a user. Registration is free. Then, login and navigate to the UPLOAD tab, where you will find detailed instructions on how to use the tool. If you encounter any issues or have any questions when using PACE, please email PLOS at figures@plos.org. Please note that Supporting Information files do not need this step.<quillbot-extension-portal></quillbot-extension-portal>

---

## [Author Response · Author response to Decision Letter 0]

7 Jul 2023

Response to reviewers is in an attached Word file that was uploaded with the revised manuscript.

---

## [Decision Letter · Decision Letter 1]

10 Aug 2023

Dose-dependent effects of rumen-protected choline on hepatic metabolism during induction of fatty liver in dry pregnant dairy cows

PONE-D-23-11577R1

Dear Dr. Santos,

We’re pleased to inform you that your manuscript has been judged scientifically suitable for publication and will be formally accepted for publication once it meets all outstanding technical requirements.

Kind regards,

Fei Luo

Academic Editor

PLOS ONE

Additional Editor Comments (optional):

Reviewers' comments:

Reviewer's Responses to Questions

**Comments to the Author**

1. If the authors have adequately addressed your comments raised in a previous round of review and you feel that this manuscript is now acceptable for publication, you may indicate that here to bypass the “Comments to the Author” section, enter your conflict of interest statement in the “Confidential to Editor” section, and submit your "Accept" recommendation.

Reviewer #1: All comments have been addressed

Reviewer #2: All comments have been addressed

2. Is the manuscript technically sound, and do the data support the conclusions?

Reviewer #1: Yes

Reviewer #2: Yes

3. Has the statistical analysis been performed appropriately and rigorously? 

Reviewer #1: Yes

Reviewer #2: Yes

4. Have the authors made all data underlying the findings in their manuscript fully available?

Reviewer #1: Yes

Reviewer #2: Yes

5. Is the manuscript presented in an intelligible fashion and written in standard English?

Reviewer #1: Yes

Reviewer #2: Yes

6. Review Comments to the Author

Reviewer #1: The authors did a thorough and comprehensive revision addressing all points raised and provided information and discussion about several aspects in their response letter which was interesting to read. This is highly appreciated. I have no further comments to add.

Reviewer #2: The article is a nice contribution to the understanding of choline action on cows with negative energy balance. It is suitable for publication.

7. PLOS authors have the option to publish the peer review history of their article (what does this mean?). If published, this will include your full peer review and any attached files.

Reviewer #1: No

Reviewer #2: No

---

## [Editor Report · Acceptance letter]

26 Sep 2023

PONE-D-23-11577R1 

Dose-dependent effects of rumen-protected choline on hepatic metabolism during induction of fatty liver in dry pregnant dairy cows 

Dear Dr. Santos:

I'm pleased to inform you that your manuscript has been deemed suitable for publication in PLOS ONE. Congratulations! Your manuscript is now with our production department. 

Kind regards, 

on behalf of

Professor Fei Luo 

Academic Editor

PLOS ONE